# META-LEARNING IN GAMES

**Keegan Harris**[*]
Carnegie Mellon University
keeganh@cs.cmu.edu

**Ioannis Anagnostides**[*]
Carnegie Mellon University
ianagnos@cs.cmu.edu

**Gabriele Farina**
FAIR, Meta AI
gfarina@meta.com

**Mikhail Khodak**
Carnegie Mellon University
mkhodak@cs.cmu.edu

**Zhiwei Steven Wu**
Carnegie Mellon University
zhiweiw@cs.cmu.edu

**Tuomas Sandholm**
Carnegie Mellon University
Strategy Robot, Inc.
Optimized Markets, Inc.
Strategic Machine, Inc.
sandholm@cs.cmu.edu

## ABSTRACT

In the literature on game-theoretic equilibrium finding, focus has mainly been on solving a single game in isolation. In practice, however, strategic interactions—ranging from routing problems to online advertising auctions—evolve dynamically, thereby leading to many similar games to be solved. To address this gap, we introduce *meta-learning* for equilibrium finding and learning to play games. We establish the first meta-learning guarantees for a variety of fundamental and well-studied classes of games, including two-player zero-sum games, general-sum games, and Stackelberg games. In particular, we obtain rates of convergence to different game-theoretic equilibria that depend on natural notions of similarity between the sequence of games encountered, while at the same time recovering the known single-game guarantees when the sequence of games is arbitrary. Along the way, we prove a number of new results in the single-game regime through a simple and unified framework, which may be of independent interest. Finally, we evaluate our meta-learning algorithms on endgames faced by the poker agent *Libratus* against top human professionals. The experiments show that games with varying stack sizes can be solved significantly faster using our meta-learning techniques than by solving them separately, often by an order of magnitude.

## 1 INTRODUCTION

Research on game-theoretic equilibrium computation has primarily focused on solving a single game in isolation. In practice, however, there are often many similar games which need to be solved. One use-case is the setting where one wants to find an equilibrium for each of multiple game variations—for example poker games where the players have various sizes of chip stacks. Another use-case is strategic interactions that evolve dynamically: in online advertising auctions, the advertiser's value for different keywords adapts based on current marketing trends (Nekipelov et al., 2015); routing games—be it Internet routing or physical transportation—reshape depending on the topology and the cost functions of the underlying network (Hoefer et al., 2011); and resource allocation problems (Johari and Tsitsiklis, 2004) vary based on the values of the goods/services. Successful agents in such complex decentralized environments must effectively learn how to incorporate past experience from previous strategic interactions in order to adapt their behavior to the current and future tasks.

*Meta-learning*, or *learning-to-learn* (Thrun and Pratt, 1998), is a common formalization for machine learning in dynamic single-agent environments. In the meta-learning framework, a learning agent faces a sequence of tasks, and the goal is to use knowledge gained from previous tasks in order to improve performance on the current task at hand. Despite rapid progress in this line of work, prior results have not been tailored to tackle multiagent settings. This begs the question: *Can players obtain provable performance improvements when meta-learning across a sequence of games?* We answer this

---

[*]Equal contribution.

question in the affirmative by introducing meta-learning for equilibrium finding and learning to play games, and providing the first performance guarantees in a number of fundamental multiagent settings.

## 1.1 Overview of Our Results

Our main contribution is to develop a general framework for establishing the first provable guarantees for meta-learning in games, leading to a comprehensive set of results in a variety of well-studied multiagent settings. In particular, our results encompass environments ranging from two-player zero-sum games with general constraint sets (and multiple extensions thereof), to general-sum games and Stackelberg games. See Table 1 for a summary of our results. Our refined guarantees are parameterized based on natural similarity metrics between the sequence of games. For example, in zero-sum games we obtain last-iterate rates that depend on the variance of the Nash equilibria (Theorem 3.2); in potential games based on the deviation of the potential functions (Theorem 3.4); and in Stackelberg games our regret bounds depend on the similarity of the leader's optimal commitment in hindsight (Theorem 3.8). All of these measures are algorithm-independent, and tie naturally to the underlying game-theoretic solution concepts.

Importantly, our algorithms are agnostic to how similar the games are, but are nonetheless specifically designed to adapt to the similarity. Our guarantees apply under a broad class of no-regret learning algorithms, such as *optimistic mirror descent (OMD)* (Chiang et al., 2012; Rakhlin and Sridharan, 2013b), with the important twist that each player employs an additional regret minimizer for meta-learning the parameterization of the base-learner; the latter component builds on the meta-learning framework of Khodak et al. (2019). For example, in zero-sum games we leverage an initialization-dependent *RVU bound* (Syrgkanis et al., 2015) in order to meta-learn the initialization of OMD across the sequences of games, leading to per-game convergence rates to Nash equilibria that closely match our refined lower bound (Theorem 3.3). More broadly, in the worst-case—*i.e.*, when the sequence of games is arbitrary— we recover the near-optimal guarantees known for static games, but as the similarity metrics become more favorable we establish significant gains in terms of convergence to different notions of equilibria.

Along the way, we also obtain new insights and results even from a single-game perspective, including convergence rates of OMD and the *extra-gradient method* in Hölder continuous variational inequalities (Rakhlin and Sridharan, 2013a), and certain nonconvex-nonconcave problems such as those considered by (Diakonikolas et al., 2021) and stochastic games. Further, our analysis is considerably simpler than prior techniques and unifies several prior results. Finally, in Section 4 we evaluate our techniques on a series of poker endgames faced by the poker agent *Libratus* (Brown and Sandholm, 2018) against top human professionals. The experiments show that our meta-learning algorithms offer significant gains compared to solving each game in isolation, often by an order of magnitude.

Table 1: A summary of our key theoretical results on meta-learning in games.

| Class of games | Specific problem | Key results | Location |
|---|---|---|---|
| Zero-sum games | Bilinear saddle-point problems | Theorems 3.1 and 3.2 | Section 3.1 |
| | Hölder continuous VIs | Theorems C.17 and C.34 | Appendices C.2 and C.6 |
| | Lower bound | Theorem 3.3 | Section 3.1 |
| General-sum games | Potential games | Theorem 3.4 | Section 3.2 |
| | (Coarse) Correlated equilibria | Theorems D.7 and D.10 | Appendices D.2 and D.3 |
| | Approximately optimal welfare | Theorem 3.6 | Section 3.2 |
| Stackelberg games | Security games | Theorem 3.8 | Section 3.3 |

## 1.2 Related Work

While most prior work on learning in games posits that the underlying game remains invariant, there is ample motivation for studying games that gradually change over time, such as online advertising (Nekipelov et al., 2015; Lykouris et al., 2016; Nisan and Noti, 2017) or congestion games (Hoefer et al., 2011; Bertrand et al., 2020; Meigs et al., 2017). Indeed, a number of prior works study the performance of learning algorithms in time-varying zero-sum games (Zhang et al., 2022b; Fiez et al., 2021b; Duvocelle et al., 2022; Cardoso et al., 2019); there, it is natural to espouse dynamic notions of regret (Yang et al., 2016; Zhao et al., 2020). A work closely related to ours is the recent paper by Zhang et al. (2022b), which provides regret bounds in time-varying bilinear saddle-point problems parameter-

ized by the similarity of the payoff matrices and the equilibria of those games. In contrast to our meta-learning setup, they study a more general setting in which the game can change arbitrarily from round-to-round. While our problem can be viewed a special type of a time-varying game in which the boundaries between different games are fixed and known, algorithms designed for generic time-varying games will not perform as well in our setting, as they do not utilize this extra information. As a result, we view these results as complementary to ours. For a more detailed discussion, see Appendix A.

An emerging paradigm for modeling such considerations is meta-learning, which has gained increasing popularity in the machine learning community in recent years; for a highly incomplete set of pointers, we refer to (Balcan et al., 2015b; Al-Shedivat et al., 2018; Finn et al., 2017; 2019; Balcan et al., 2019; Li et al., 2017; Chen et al., 2022), and references therein. Our work constitutes the natural coalescence of meta-learning with the line of work on (decentralized) online learning in games. Although, as we pointed out earlier, learning in dynamic games has already received considerable attention, we are the first (to our knowledge) to formulate and address such questions within the meta-learning framework; *c.f.*, see (Kayaalp et al., 2020; 2021; Li et al., 2022). Finally, our methods may be viewed within the *algorithms with predictions* paradigm (Mitzenmacher and Vassilvitskii, 2020): we speed up equilibrium computation by learning to predict equilibria across multiple games, with the task-similarity being the measure of prediction quality. For further related work, see Appendix A.

## 2 OUR SETUP: META-LEARNING IN GAMES

**Notation** We use boldface symbols to represent vectors and matrices. Subscripts are typically reserved to indicate the player, while superscripts usually correspond to the iteration or the index of the task. We let $\mathbb{N} := \{1, 2, \dots, \}$ be the set of natural numbers. For $T \in \mathbb{N}$, we use the shorthand notation $[\![T]\!] := \{1, 2, \dots, T\}$. For a nonempty convex and compact set $\mathcal{X}$, we denote by $\Omega_{\mathcal{X}}$ its $\ell_2$-diameter: $\Omega_{\mathcal{X}} := \max_{\boldsymbol{x}, \boldsymbol{x}' \in \mathcal{X}} \|\boldsymbol{x} - \boldsymbol{x}'\|_2$. Finally, to lighten the exposition we use the $O(\cdot)$ notation to suppress factors that depend polynomially on the natural parameters of the problem.

**The general setup** We consider a setting wherein players interact in a sequence of $T$ repeated games (or *tasks*), for some $\mathbb{N} \ni T \gg 1$. Each task itself consists of $m \in \mathbb{N}$ iterations. Any fixed task $t$ corresponds to a multiplayer game $\mathcal{G}^{(t)}$ between a set $[\![n]\!]$ of players, with $n \geq 2$; it is assumed for simplicity in the exposition that $n$ remains invariant across the games, but some of our results apply more broadly. Each player $k \in [\![n]\!]$ selects a strategy $\boldsymbol{x}_k$ from a convex and compact set of strategies $\mathcal{X}_k \subseteq \mathbb{R}^{d_k}$ with nonempty relative interior. For a given joint strategy profile $\boldsymbol{x} := (\boldsymbol{x}_1, \dots, \boldsymbol{x}_n) \in \bigtimes_{k=1}^{n} \mathcal{X}_k$, there is a multilinear utility function $u_k : \boldsymbol{x} \mapsto \langle \boldsymbol{x}_k, \boldsymbol{u}_k(\boldsymbol{x}_{-k}) \rangle$ for each player $k$, where $\boldsymbol{x}_{-k} := (\boldsymbol{x}_1, \dots, \boldsymbol{x}_{k-1}, \boldsymbol{x}_{k+1}, \dots, \boldsymbol{x}_n)$. We will also let $L > 0$ be a Lipschitz parameter of each game, in the sense that for any player $k \in [\![n]\!]$ and any two strategy profiles $\boldsymbol{x}_{-k}, \boldsymbol{x}'_{-k} \in \bigtimes_{k' \neq k} \mathcal{X}_{k'}$,

$$\|\boldsymbol{u}_k(\boldsymbol{x}_{-k}) - \boldsymbol{u}_k(\boldsymbol{x}'_{-k})\|_2 \leq L \|\boldsymbol{x}_{-k} - \boldsymbol{x}'_{-k}\|_2. \tag{1}$$

Here, we use the $\ell_2$-norm for convenience in the analysis; (1) can be translated to any equivalent norm. Finally, for a joint strategy profile $\boldsymbol{x} \in \bigtimes_{k=1}^{n} \mathcal{X}_k$, the *social welfare* is defined as $\mathrm{SW}(\boldsymbol{x}) := \sum_{k=1}^{n} u_k(\boldsymbol{x})$, so that $\mathrm{OPT} := \max_{\boldsymbol{x} \in \bigtimes_{k=1}^{n} \mathcal{X}_k} \mathrm{SW}(\boldsymbol{x})$ denotes the optimal social welfare.

A concrete example encompassed by our setup is that of *extensive-form games*. More broadly, it captures general games with concave utilities (Rosen, 1965; Hsieh et al., 2021).

**Online learning in games** Learning proceeds in an online fashion as follows. At every iteration $i \in [\![m]\!]$ of some underlying game $t$, each player $k \in [\![n]\!]$ has to select a strategy $\boldsymbol{x}_k^{(t,i)} \in \mathcal{X}_k$. Then, in the full information setting, the player observes as feedback the utility corresponding to the other players' strategies at iteration $i$; namely, $\boldsymbol{u}_k^{(t,i)} := \boldsymbol{u}_k(\boldsymbol{x}_{-k}^{(t,i)}) \in \mathbb{R}^{d_k}$. For convenience, we will assume that $\|\boldsymbol{u}_k(\boldsymbol{x}_{-k}^{(t,i)})\|_\infty \leq 1$. The canonical measure of performance in online learning is that of *external regret*, comparing the performance of the learner with that of the optimal fixed strategy in hindsight:

**Definition 2.1** (Regret). *Fix a player $k \in [\![n]\!]$ and some game $t \in [\![T]\!]$. The (external) regret of player $k$ is defined as*

$$\mathrm{Reg}_k^{(t,m)} := \max_{\mathring{\boldsymbol{x}}_k^{(t)} \in \mathcal{X}_k} \left\{ \sum_{i=1}^{m} \langle \mathring{\boldsymbol{x}}_k^{(t)}, \boldsymbol{u}_k^{(t,i)} \rangle \right\} - \langle \boldsymbol{x}_k^{(t,i)}, \boldsymbol{u}_k^{(t,i)} \rangle.$$

We will let $\mathring{\boldsymbol{x}}_k^{(t)}$ be an optimum-in-hindsight strategy for player $k$ in game $t$; ties are broken arbitrarily, but according to a fixed rule (*e.g.*, lexicographically). In the meta-learning setting, our goal will be to optimize the average performance—typically measured in terms of convergence to different game-theoretic equilibria—across the sequence of games.

**Optimistic mirror descent**    Suppose that $\mathcal{R}_k : \mathcal{X}_k \to \mathbb{R}$ is a 1-strongly convex regularizer with respect to a norm $\|\cdot\|$. We let $\mathcal{B}_{\mathcal{R}_k}(\boldsymbol{x}_k \parallel \boldsymbol{x}_k') \coloneqq \mathcal{R}_k(\boldsymbol{x}_k) - \mathcal{R}_k(\boldsymbol{x}_k') - \langle \nabla \mathcal{R}_k(\boldsymbol{x}_k'), \boldsymbol{x}_k - \boldsymbol{x}_k' \rangle$ denote the *Bregman divergence* induced by $\mathcal{R}_k$, where $\boldsymbol{x}_k'$ is in the relative interior of $\mathcal{X}_k$. *Optimistic mirror descent (OMD)* (Chiang et al., 2012; Rakhlin and Sridharan, 2013b) is parameterized by a prediction $\boldsymbol{m}_k^{(t,i)} \in \mathbb{R}^{d_k}$ and a learning rate $\eta > 0$, and is defined at every iteration $i \in \mathbb{N}$ as follows.

$$\boldsymbol{x}_k^{(t,i)} \coloneqq \arg \max_{\boldsymbol{x}_k \in \mathcal{X}_k} \left\{ \langle \boldsymbol{x}_k, \boldsymbol{m}_k^{(t,i)} \rangle - \frac{1}{\eta} \mathcal{B}_{\mathcal{R}_k}(\boldsymbol{x}_k \parallel \hat{\boldsymbol{x}}_k^{(t,i-1)}) \right\},$$

$$\hat{\boldsymbol{x}}_k^{(t,i)} \coloneqq \arg \max_{\hat{\boldsymbol{x}}_k \in \mathcal{X}_k} \left\{ \langle \hat{\boldsymbol{x}}_k, \boldsymbol{u}_k^{(t,i)} \rangle - \frac{1}{\eta} \mathcal{B}_{\mathcal{R}_k}(\hat{\boldsymbol{x}}_k \parallel \hat{\boldsymbol{x}}_k^{(t,i-1)}) \rangle \right\}.$$

Further, $\hat{\boldsymbol{x}}_k^{(1,0)} \coloneqq \arg \min_{\hat{\boldsymbol{x}}_k \in \mathcal{X}_k} \mathcal{R}_k(\hat{\boldsymbol{x}}_k) =: \boldsymbol{x}_k^{(1,0)}$, and $\boldsymbol{m}_k^{(t,1)} \coloneqq \boldsymbol{u}_k(\boldsymbol{x}_{-k}^{(t,0)})$. Under Euclidean regularization, $\mathcal{R}_k(\boldsymbol{x}_k) \coloneqq \frac{1}{2}\|\boldsymbol{x}_k\|_2^2$, we will refer to OMD as *optimistic gradient descent (OGD)*.

## 3    META-LEARNING HOW TO PLAY GAMES

In this section, we present our main theoretical results: provable guarantees for online and decentralized meta-learning in games. We commence with zero-sum games in Section 3.1, and we then transition to general-sum games (Section 3.2) and Stackelberg (security) games (Section 3.3).

### 3.1    ZERO-SUM GAMES

We first highlight our results for bilinear saddle-point problems (BSPPs), which take the form $\min_{\boldsymbol{x} \in \mathcal{X}} \max_{\boldsymbol{y} \in \mathcal{Y}} \boldsymbol{x}^\top \mathbf{A} \boldsymbol{y}$, where $\mathbf{A} \in \mathbb{R}^{d_x \times d_y}$ is the payoff matrix of the game. A canonical application for this setting is on the solution of zero-sum imperfect-information extensive-form games (Romanovskii, 1962; Koller and Megiddo, 1992), as we explore in our experiments (Section 4). Next we describe a number of extensions to gradually more general settings, and we conclude with our lower bound (Theorem 3.3). The proofs from this subsection are included in Appendix C.

We first derive a refined meta-learning convergence guarantee for the average of the players' strategies. Below, we denote by $V_x^2 \coloneqq \frac{1}{T} \min_{\boldsymbol{x} \in \mathcal{X}} \sum_{t=1}^T \|\mathring{\boldsymbol{x}}^{(t)} - \boldsymbol{x}\|_2^2$ the task similarity metric for player $x$, written in terms of the optimum-in-hindsight strategies; analogous notation is used for player $y$.

**Theorem 3.1** (Informal; Detailed Version in Corollary C.2). *Suppose that both players employ* OGD *with a suitable (fixed) learning rate and follow the leader over previous optimum-in-hindsight strategies for the initialization. Then, the game-average duality gap of the players' average strategies is bounded by*

$$\frac{1}{T} \sum_{t=1}^T \frac{1}{m} \left( \mathrm{Reg}_x^{(t,m)} + \mathrm{Reg}_y^{(t,m)} \right) \leq \frac{2L}{m} \left( V_x^2 + V_y^2 \right) + \frac{8L(1 + \log T)}{mT} \left( \Omega_{\mathcal{X}}^2 + \Omega_{\mathcal{Y}}^2 \right). \quad (2)$$

Here, the second term in the right-hand side of (2) becomes negligible for a large number of games $T$, while the first term depends on the task similarity measures. For any sequence of games, Theorem 3.1 nearly matches the lower bound in the single-task setting (Daskalakis et al., 2015), but our guarantee can be significantly better when $V_x^2, V_y^2 \ll 1$. To achieve this, the basic idea is to use—on top of OGD—a "meta" regret minimization algorithm that, for each player, learns a sequence of initializations by taking the average of the past optima-in-hindsight, which is equivalent to *follow the leader (FTL)* over the regret upper-bounds of the within-task algorithm; see Algorithm 1 (in Appendix B) for pseudocode of the meta-version of OGD we consider. Similar results can be obtained more broadly for OMD (*c.f.*, see Appendices D.2 and D.3). We also obtain analogous refined bounds for the *individual* regret of each player (Corollary C.4).

One caveat of Theorem 3.1 is that the underlying task similarity measure could be algorithm-dependent, as the optimum-in-hindsight for each player could depend on the other player's

behavior. To address this, we show that if the meta-learner can initialize using *Nash equilibria (NE)* (recall Definition C.5) from previously seen games, the game-average last-iterate rates gracefully decrease with the similarity of the Nash equilibria of those games. More precisely, if $\boldsymbol{z} := (\boldsymbol{x}, \boldsymbol{y}) \in \mathcal{X} \times \mathcal{Y} =: \mathcal{Z}$, we let $V_{\text{NE}}^2 := \frac{1}{T} \max_{\boldsymbol{z}^{(1,\star)}, \ldots, \boldsymbol{z}^{(T,\star)}} \min_{\boldsymbol{z} \in \mathcal{Z}} \sum_{t=1}^{T} \|\boldsymbol{z}^{(t,\star)} - \boldsymbol{z}\|_2^2$, where $\boldsymbol{z}^{(t,\star)}$ is any Nash equilibrium of the $t$-th game. As we point out in the sequel, we also obtain results under a more favorable notion of task similarity that does not depend on the worst sequence of NE.

**Theorem 3.2** (Informal; Detailed Version in Theorem C.8). *When both players employ* `OGD` *with a suitable (fixed) learning rate and* `FTL` *over previous NE strategies for the initialization, then*

$$\bar{m} \le \frac{2 V_{NE}^2}{\epsilon^2} + \frac{8(1 + \log T)}{T \epsilon^2} \left( \Omega_{\mathcal{X}}^2 + \Omega_{\mathcal{Y}}^2 \right)$$

*iterations suffice to reach an $O(\epsilon)$-approximate Nash equilibrium on average across the $T$ games.*

Theorem 3.2 recovers the optimal $m^{-1/2}$ rates for `OGD` (Golowich et al., 2020a;b) under an arbitrary sequence of games, but offers substantial gains in terms of the average iteration complexity when the Nash equilibria of the games are close. For example, when they lie within a ball of $\ell_2$-diameter $\sqrt{\delta(\Omega_{\mathcal{X}}^2 + \Omega_{\mathcal{Y}}^2)}$, for some $\delta \in (0, 1]$, Theorem 3.2 improves upon the rate of `OGD` by at least a multiplicative factor of $1/\delta$ as $T \to \infty$. While *generic*—roughly speaking, randomly perturbed—zero-sum (normal-form) games have a unique Nash equilibrium (van Damme, 1987), the worst-case NE similarity metric used in Theorem 3.2 can be loose under multiplicity of equilibria. For that reason, in Appendix C.1.2 we further refine Theorem 3.2 using the most favorable sequence of Nash equilibria; this requires that players know each game after its termination, which is arguably a well-motivated assumption in some applications. We further remark that Theorem 3.2 can be cast in terms of the similarity $V_x^2 + V_y^2$, instead of $V_{\text{NE}}^2$, using the parameterization of Theorem 3.1. Finally, since the base-learner can be viewed as an algorithm with predictions—the number of iterations to compute an approximate NE is smaller if the Euclidean error of a prediction of it (the initialization) is small—Theorem 3.2 can also be viewed as *learning* these predictions (Khodak et al., 2022) by targeting that error measure.

**Extensions** Moving beyond bilinear saddle-point problems, we extend our results to gradually broader settings. First, in Appendix C.2 we apply our techniques to general variational inequality problems under a Lipschitz continuous operator for which the so-called *MVI property* (Mertikopoulos et al., 2019) holds. Thus, Theorems 3.1 and 3.2 are extended to settings such as smooth convex-concave games and zero-sum polymatrix (multiplayer) games (Cai et al., 2016). Interestingly, extensions are possible even under the *weak MVI property* (Diakonikolas et al., 2021), which captures certain "structured" nonconvex-nonconcave games. In a similar vein, we also study the challenging setting of Shapley's stochastic games (Shapley, 1953) (Appendix C.5). There, we show that there exists a time-varying—instead of constant—but non-vanishing learning rate schedule for which `OGD` reaches minimax equilibria, thereby leading to similar extensions in the meta-learning setting. Next, we relax the underlying Lipschitz continuity assumption underpinning the previous results by instead imposing only $\alpha$-Hölder continuity (recall Definition C.32). We show that in such settings `OGD` enjoys a rate of $m^{-\alpha/2}$ (Theorem C.34), which is to our knowledge a new result; in the special case where $\alpha = 1$, we recover the recently established $m^{-1/2}$ rates. Finally, while we have focused on the `OGD` algorithm, our techniques apply to other learning dynamics as well. For example, in Appendix C.7 we show that the extensively studied extra-gradient (`EG`) algorithm (Korpelevich, 1976) can be analyzed in a unifying way with `OGD`, thereby inheriting all of the aforementioned results under `OGD`; this significantly broadens the implications of (Mokhtari et al., 2020), which only applied in certain unconstrained problems. Perhaps surprisingly, although `EG` is *not* a no-regret algorithm, our analysis employs a regret-based framework using a suitable proxy for the regret (see Theorem C.35).

**Lower bound** We conclude this subsection with a lower bound, showing that our guarantee in Theorem 3.1 is essentially sharp under a broad range of our similarity measures. Our result significantly refines the single-game lower bound of Daskalakis et al. (2015) by constructing an appropriate distribution over sequences of zero-sum games.

**Theorem 3.3** (Informal; Precise Version in Theorem C.39). *For any $\epsilon > 0$, there exists a distribution over sequences of $T$ zero-sum games, with a sufficiently large $T = T(\epsilon)$, such that*

$$\frac{1}{T} \sum_{t=1}^{T} \mathbb{E}[\text{Reg}_x^{(t,m)} + \text{Reg}_y^{(t,m)}] \ge \frac{1}{2} \left( V_x^2 + V_y^2 \right) - \epsilon = \frac{1}{2} V_{NE}^2 - \epsilon.$$

## 3.2 GENERAL-SUM GAMES

In this subsection, we switch our attention to general-sum games. Here, unlike zero-sum games, no-regret learning algorithms are instead known to generally converge—in a time-average sense—to *correlated equilibrium* concepts, which are more permissive than the Nash equilibrium. Nevertheless, there are structured classes of general-sum games for which suitable dynamics do reach Nash equilibria; perhaps the most notable example being that of *potential games*. In this context, we first obtain meta-learning guarantees for potential games, parameterized by the similarity of the potential functions. Then, we derive meta-learning algorithms with improved guarantees for convergence to correlated and *coarse* correlated equilibria. Finally, we conclude this subsection with improved guarantees of convergence to near-optimal—in terms of social welfare—equilibria. Proofs from this subsection are included in Appendices B and D.

**Potential games** A potential game is endowed with the additional property of admitting a potential: a player-independent function that captures the player's benefit from unilaterally deviating from any given strategy profile (Definition D.2). In our meta-learning setting, we posit a sequence of potential games $(\Phi^{(t)})_{1 \leq t \leq t}$, each described by its potential function. Unlike our approach in Section 3.1, a central challenge here is that the potential function is in general nonconcave/nonconvex, precluding standard regret minimization approaches. Instead, we find that by initializing at the previous last-iterate the dynamics still manage to adapt based on the similarity $V_\Delta := \frac{1}{T} \sum_{t=1}^{T-1} \Delta(\Phi^{(t)}, \Phi^{(t+1)})$, where $\Delta(\Phi, \Phi') := \max_{\boldsymbol{x}}(\Phi(\boldsymbol{x}) - \Phi'(\boldsymbol{x}))$, which captures the deviation of the potential functions. This initialization has the additional benefit of being agnostic to the boundaries of different tasks. Unlike our results in Section 3.1, the following guarantee applies even for vanilla (*i.e.*, non-optimistic) projected gradient descent (GD).

**Theorem 3.4** (Informal; Detailed Version in Corollary D.5). *GD with suitable parameterization requires $O\left(\frac{V_\Delta}{\epsilon^2} + \frac{\Phi_{max}}{\epsilon^2 T}\right)$ iterations to reach an $\epsilon$-approximate Nash equilibrium on average across the $T$ potential games, where $\max_{\boldsymbol{x},t} |\Phi^{(t)}(\boldsymbol{x})| \leq \Phi_{max}$.*

Theorem 3.4 matches the known rate of GD for potential games in the worst case, but offers substantial gains in terms of the average iteration complexity when the games are similar. For example, if $|\Phi^{(t)}(\boldsymbol{x}) - \Phi^{(t-1)}(\boldsymbol{x})| \leq \alpha$, for all $\boldsymbol{x} \in \times_{k=1}^{n} \mathcal{X}_k$ and $t \geq 2$, then $O(\alpha/\epsilon^2)$ iterations suffice to reach an $\epsilon$-approximate NE on an average game, as $T \to +\infty$. Such a scenario may arise in, *e.g.*, a sequence of routing games if the cost functions for each edge change only slightly between games.

**Convergence to correlated equilibria** In contrast, for general games the best one can hope for is to obtain improved rates for convergence to correlated or coarse correlated equilibria (Hart and Mas-Colell, 2000; Blum and Mansour, 2007). It is important to stress that learning correlated equilibria is fundamentally different than learning Nash equilibria—which are product distributions. For example, for the former any initialization—which is inevitably a product distribution in the case of uncoupled dynamics—could fail to exploit the learning in the previous task (Proposition D.1): unlike Nash equilibria, correlated equilibria (in general) cannot be decomposed for each player, thereby making uncoupled methods unlikely to adapt to the similarity of CE. Instead, our task similarity metrics depend on the optima-in-hindsight for each player. Under this notion of task similarity, we obtain task-average guarantees for CCE by meta-learning the initialization (by running FTL) and the learning rate (by running the EWOO method of Hazan et al. (2007) over a sequence of regret upper bounds) of *optimistic hedge* (Daskalakis et al., 2021) (Theorem D.7)—OMD with entropic regularization. Similarly, to obtain guarantees for CE, we use the *no-swap-regret* construction of Blum and Mansour (2007) in conjuction with the logarithmic barrier (Anagnostides et al., 2022a) (Theorem D.10).

### 3.2.1 SOCIAL WELFARE GUARANTEES

We conclude this subsection with meta-learning guarantees for converging to near-optimal equilibria (Theorem 3.6). Let us first recall the following central definition.

**Definition 3.5** (Smooth games (Roughgarden, 2015)). *A game $\mathcal{G}$ is $(\lambda, \mu)$-smooth, with $\lambda, \mu > 0$, if there exists a strategy profile $\boldsymbol{x}^\star \in \times_{k=1}^{n} \mathcal{X}_k$ such that for any $\boldsymbol{x} \in \times_{k=1}^{n} \mathcal{X}_k$,*

$$\sum_{k=1}^{n} u_k(\boldsymbol{x}_k^\star, \boldsymbol{x}_{-k}) \geq \lambda \text{OPT} - \mu \text{SW}(\boldsymbol{x}), \tag{3}$$

*where* OPT *is the optimal social welfare and* SW($\boldsymbol{x}$) *is the social welfare of joint strategy profile* $\boldsymbol{x}$.

Smooth games capture a number of important applications, including network congestion games (Awerbuch et al., 2013; Christodoulou and Koutsoupias, 2005) and simultaneous auctions (Christodoulou et al., 2016; Roughgarden et al., 2017) (see Appendix B for additional examples); both of those settings are oftentimes non-static in real-world applications, thereby motivating our meta-learning considerations. In this context, we assume that there is a sequence of smooth games $(\mathcal{G}^{(t)})_{1 \le t \le T}$, each of which is $(\lambda^{(t)}, \mu^{(t)})$-smooth (Definition 3.5).

**Theorem 3.6** (Informal; Detailed Version in Theorem B.11). *If all players use* OGD *with suitable parameterization in a sequence of* $T$ *games* $(\mathcal{G}^{(t)})_{1 \le t \le T}$, *each of which is* $(\lambda^{(t)}, \mu^{(t)})$-*smooth, then*

$$\frac{1}{mT} \sum_{t=1}^{T} \sum_{i=1}^{m} \text{SW}(\boldsymbol{x}^{(t,i)}) \ge \frac{1}{T} \sum_{t=1}^{T} \frac{\lambda^{(t)}}{1 + \mu^{(t)}} \text{OPT}^{(t)} - \frac{2L\sqrt{n-1}}{m} \sum_{k=1}^{n} V_k^2 - \widetilde{O}\left(\frac{1}{mT}\right), \quad (4)$$

*where* OPT$^{(t)}$ *is the optimal social welfare attainable at game* $\mathcal{G}^{(t)}$ *and* $\widetilde{O}(\cdot)$ *hides logarithmic terms.*

The first term in the right-hand side of (4) is the average robust PoA in the sequence of games, while the third term vanishes as $T \to \infty$. The orchestrated learning dynamics reach approximately optimal equilibria much faster when the underlying task similarity is small; without meta-learning one would instead obtain the $m^{-1}$ rate known from the work of Syrgkanis et al. (2015). Theorem 3.6 is established by first providing a refined guarantee for the *sum of the players regrets* (Theorem B.3), and then translating that guarantee in terms of the social welfare using the smoothness condition for each game (Proposition B.10). Our guarantees are in fact more general, and apply for any suitable linear combination of players' utilities (see Corollary B.12).

## 3.3 STACKELBERG (SECURITY) GAMES

To conclude our theoretical results, we study meta-learning in repeated Stackelberg games. Following the convention of Balcan et al. (2015a), we present our results in terms of Stackelberg security games, although our results apply to general Stackelberg games as well (see (Balcan et al., 2015a, Section 8) for details on how such results extend).

**Stackelberg security games** A repeated Stackelberg security game is a sequential interaction between a defender and $m$ attackers. In each round, the defender commits to a mixed strategy over $d$ targets to protect, which induces a *coverage probability vector* $\boldsymbol{x} \in \Delta^d$ over targets. After having observed coverage probability vector, the attacker *best responds* by attacking some target $b(\boldsymbol{x}) \in [\![d]\!]$ in order to maximize their utility in expectation. Finally, the defender's utility is some function of their coverage probability vector $\boldsymbol{x}$ and the target attacked $b(\boldsymbol{x})$.

It is a well-known fact that no-regret learning in repeated Stackelberg games is not possible without any prior knowledge about the sequence of followers (Balcan et al., 2015a, Section 7), so we study the setting in which each attacker belongs to one of $k$ possible *attacker types*. We allow sequence of attackers to be adversarially chosen from the $k$ types, and assume the attacker's type is revealed to the leader after each round. We adapt the methodology of Balcan et al. (2015a) to our setting by meta-learning the initialization and learning rate of the multiplicative weights update (henceforth MWU) run over a finite (but exponentially-large) set of *extreme points* $\mathcal{E} \subset \Delta^d$.[1] Each point $\boldsymbol{x} \in \mathcal{E}$ corresponds to a leader mixed strategy, and $\mathcal{E}$ can be constructed in such a way that it will always contain a mixed strategy which is arbitrarily close to the optima-in-hindsight for each task.[2]

Our results are given in terms of guarantees on the task-average *Stackelberg regret*, which measures the difference in utility between the defender's deployed sequence of mixed strategies and the optima-in-hindsight, given that the attacker best responds.

**Definition 3.7** (Stackelberg Regret). *Denote attacker* $f^{(t,i)}$'*s best response to mixed strategy* $\boldsymbol{x}$ *as* $b_{f^{(t,i)}}(\boldsymbol{x})$. *The Stackelberg regret of the attacker in a repeated Stackelberg security game* $t$ *is*

$$\text{StackReg}^{(t,m)}(\mathring{\boldsymbol{x}}^{(t)}) = \sum_{i=1}^{m} \langle \mathring{\boldsymbol{x}}^{(t)}, \boldsymbol{u}^{(t)}(b_{f^{(t,i)}}(\mathring{\boldsymbol{x}}^{(t)})) \rangle - \langle \boldsymbol{x}^{(t,i)}, \boldsymbol{u}^{(t)}(b_{f^{(t,i)}}(\boldsymbol{x}^{(t,i)})) \rangle.$$

---

[1] This is likely unavoidable, as Li et al. (2016) show computing a Stackelberg strategy is strongly NP-Hard.
[2] For a precise definition of how to construct $\mathcal{E}$, we point the reader to (Balcan et al., 2015a, Section 4).

In contrast to the standard notion of regret (Definition 2.1), Stackelberg regret takes into account the extra structure in the defender's utility in Stackelberg games; namely that it is a function of the defender's current mixed strategy (through the attacker's best response).

**Theorem 3.8** (Informal; Detailed Version in Theorem E.1). *Given a sequence of $T$ repeated Stackelberg security games with $d$ targets, $k$ attacker types, and within-game time-horizon $m$, running* MWU *over the set of extreme points $\mathcal{E}$ as defined in Balcan et al. (2015a) with suitable initialization and sequence of learning rates achieves task-averaged expected Stackelberg regret*

$$\frac{1}{T} \sum_{t=1}^{T} \mathbb{E}[\text{StackReg}^{(t,m)}] = O(\sqrt{H(\bar{\boldsymbol{y}})m}) + o_T(\text{poly}(m, |\mathcal{E}|)),$$

*where the sequence of attackers in each task can be adversarially chosen, the expectation is with respect to the randomness of* MWU, *$\bar{\boldsymbol{y}} := \frac{1}{T} \sum_{t=1}^{T} \mathring{\boldsymbol{y}}^{(t)}$, where $\mathring{\boldsymbol{y}}^{(t)}$ is the optimum-in-hindsight distribution over mixed strategies in $\mathcal{E}$ for game $t$, $H(\bar{\boldsymbol{y}})$ is the Shannon entropy of $\bar{\boldsymbol{y}}$, and $o_T(1)$ suppresses terms which decay with $T$.*

$H(\bar{\boldsymbol{y}}) \leq \log |\mathcal{E}|$, so in the worst-case our algorithm asymptotically matches the $O(\sqrt{m \log |\mathcal{E}|})$ performance of the algorithm of Balcan et al. (2015a). Entropy $H(\bar{\boldsymbol{y}})$ is small whenever the same small set of mixed strategies are optimal for the sequence of $T$ Stackelberg games. For example, if in each task the adversary chooses from $s \ll k$ attacker types who are only interested in attacking $u \ll d$ targets (unbeknownst to the meta-learner), $H(\bar{\boldsymbol{y}}) = O(s^2 u \log(su))$. In Stackelberg security games $|\mathcal{E}| = O((2^d + kd^2)^d d^k)$, so $\log |\mathcal{E}| = O(d^2 k \log(dk))$. Finally, the distance between the set of optimal strategies does not matter, as $\bar{\boldsymbol{y}}$ is a categorical distribution over a discrete set of mixed strategies.

## 4 Experiments

In this section, we evaluate our meta-learning techniques in two River endgames that occurred in the *Brains vs AI* competition (Brown and Sandholm, 2018). We use the two public endgames that were released by the authors,[3] denoted 'Endgame A' and 'Endgame B,' each corresponding to a zero-sum extensive-form game. For each of these endgames, we produced $T := 200$ individual tasks by varying the size of the stacks of each player according to three different *task sequencing setups*:[4]

1. (*random* stacks) In each task we select stack sizes for the players by sampling uniformly at random a multiple of 100 in the range $[1000, 20000]$.
2. (*sorted* stacks) Task $t \in \{1, \ldots, 200\}$ corresponds to solving the endgame where the stack sizes are set to the amount $t \times 100$ for each player.
3. (*alternating* stacks) We sequence the stack amounts of the players as follows: in task 1, the stacks are set to 100; in task 2 to $200,000$; in task 3 to 200; in task 4 to $199,900$; and so on.

For each endgame, we tested the performance when both players (1) employ OGD while meta-learning the initialization (Theorem 3.1) with $\boldsymbol{m}_x^{(t,1)} = \mathbf{0}_{d_x}$ and $\boldsymbol{m}_y^{(t,1)} = \mathbf{0}_{d_y}$, (2) employ OGD while setting the initialization equal to the last iterate of the previous task (see Remark B.8), and (3) use the vanilla initialization of OGD—*i.e.*, the players treat each game separately. For each game, players run $m := 1000$ iterations. The $\ell_2$ projection to the *sequence-form polytope* (Romanovskii, 1962; Koller and Megiddo, 1992)—the strategy set of each player in extensive-form games—required for the steps of OGD is implemented via an algorithm originally described by Gilpin et al. (2012), and further clarified in (Farina et al., 2022, Appendix B). We tried different learning rates for the players selected from the set $\{0.1, 0.01, 0.001\}$. Figure 1 illustrates our results for $\eta := 0.01$, while the others are deferred to Appendix F. In the table at the top of Figure 1 we highlight several parameters of the endgames including the board configuration, the dimensions of the players' strategy sets—*i.e.*, the sequences—and the number of nonzero elements in each payoff matrix. Because of the scale of the games, we used the *Kronecker sparsification* algorithm of Farina and Sandholm (2022, Technique A) in order to accelerate the training.

---

[3]Obtained from `https://github.com/Sandholm-Lab/LibratusEndgames`.

[4]While in the general meta-learning setup it is assumed that the number of tasks is large but per-task data is limited (*i.e.*, $T \gg m$), we found that setting $T := 200$ was already sufficient to see substantial benefits.

| Game | Board | Pot | Sequences | | Decision Points | | Payoff Matrix |
|------|-------|-----|-----------|------|-----------------|------|---------------|
| | | | Pl. 1 | Pl. 2 | Pl. 1 | Pl. 2 | num. nonzeros |
| Endgame A | J♠ K♠ 5♣ Q♠ 7♦ | 3,700 | 18,789 | 19,237 | 6,710 | 6,870 | 14,718,298 |
| Endgame B | 4♠ 8♥ 10♣ 9♥ 2♠ | 500 | 46,875 | 47,381 | 16,304 | 16,480 | 62,748,525 |

Figure 1: (Top) Parameters of each endgame. (Bottom) The task-averaged NE gap of the players' average strategies across 200 tasks, 2 endgames, and 3 different stack orderings. Both players use OGD with $\eta := 0.01$. For the random stacks, we repeated each experiment 10 times with different random seeds. The plots show the mean (thick line) as well as the minimum and maximum values. We see that across all task sequencing setups, meta-learning the initialization (using either technique) leads to up to an order of magnitude better performance compared to vanilla OGD. When stacks are sorted, initializing to the last iterate of the previous game obtains the best performance, whereas when stacks are alternated or random, initializing according to Theorem 3.1 performs best.

## 5   CONCLUSIONS AND FUTURE RESEARCH

In this paper, we introduced the study of meta-learning in games. In particular, we considered many of the most central game classes—including zero-sum games, potential games, general-sum multi-player games, and Stackelberg security games—and obtained provable performance guarantees expressed in terms of natural measures of similarity between the games. Experiments on several sequences of poker endgames that were actually played in the *Brains vs AI* competition (Brown and Sandholm, 2018) show that meta-learning the initialization improves performance even by an order of magnitude.

Our results open the door to several exciting directions for future research, including meta-learning in other settings for which single-game results are known, such as general nonconvex-nonconcave min-max problems (Suggala and Netrapalli, 2020), the nonparametric regime (Daskalakis and Golowich, 2022), and partial feedback (such as bandit) models (Wei and Luo, 2018; Hsieh et al., 2022; Balcan et al., 2022; Osadchiy et al., 2022). Another interesting, yet challenging, avenue for future research would be to consider strategy sets that can vary across tasks.

## ACKNOWLEDGEMENTS

We are grateful to the anonymous ICLR reviewers for valuable feedback. KH is supported by a NDSEG Fellowship. IA and TS are supported by NSF grants IIS-1901403 and CCF-1733556, and the ARO under award W911NF2010081. MK is supported by a Meta Research PhD Fellowship. ZSW is supported in part by the NSF grant FAI-1939606, a Google Faculty Research Award, a J.P. Morgan Faculty Award, a Meta Research Award, and a Mozilla Research Grant. The authors would like to thank Nina Balcan for helpful discussions throughout the course of the project. IA is grateful to Ioannis Panageas for insightful discussions regarding Appendix C.5.

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
