# OpenReview forum: "Meta-Learning in Games"
_ICLR.cc/2023/Conference — ICLR 2023 poster_

### Official Review · Reviewer_fvHn · 2022-10-24

**Confidence:** 3
**Correctness:** 3
**Technical Novelty And Significance:** 3
**Empirical Novelty And Significance:** 3
**Recommendation:** 6

**Clarity, Quality, Novelty And Reproducibility:**

In general, the proposed framework is considered to be novel. However, the clarity of the paper may need some improvement. In particular, as mentioned in the Weakness part, it's difficult to get the core idea of the algorithm by solely reading the main content. Meanwhile, the background for many problems it addresses can only be provided in appendix. One potential resolution is to have less discussion about results in specific problems. Another potential resolution is to consider **submitting this paper to a journal**, which has much less restrictions on the length of the main content.

### Questions
- What is the definition of average potential game (or average zero-sum game) in Theorem 3.4 (or in Theorem 3.2)? Based on the proof in appendix, it seems what actually happens is that policy for some game at some iteration reaches $O(\epsilon)$-approximate Nash equilibrium.
- It is not very clear what is the "Meta component" in the proposed algorithm, which I refer to something similar to $\mathbf{w}\_{\mathrm{MAML}}$ in "Online Meta-learning". In particular, $\mathbf{w}_{\mathrm{MAML}}$ captures the high-level similarities among different tasks. Is there anything in the algorithm or analysis that captures the similarities among different games?
- Is it possible to meta-learn these games in a batch fashion like MAML instead of an online fashion?

**Strength And Weaknesses:**

### Strengths
The results provided in this paper is very extensive, showing that the proposed framework can be readily applied to many classes of meta-learning problems related to games. Meanwhile, these results also provide connections to many different directions, opening an area of future research.

### Weaknesses

- One major weakness of this paper in my concern is in its organization. Since the coverage is too extensive, the current main content is more like a list of results. That is, it looks very difficult to get the core idea of the proposed algorithm by solely reading the main content. If my understanding is correct, most of the theoretical results are derived based on Theorem B.3. Therefore, it's probably better to put more discussion about Theorem B.3 and less discussion about specific problems into the main content. Otherwise, considering **submitting this paper to a journal** can also be an alternative choice.
- Another major weakness of this paper in my concern is in its assumptions. In particular, in this paper, the algorithm requires exact Nash equilibrium about previous games, while previous meta-learning algorithms usually don't require so much information about previous tasks (like the "Online Meta-learning"). Is there any unique properties in games making this amount of information necessary? Meanwhile, why this is a well-motivated assumptions in some applications, which seems not to be discussed in appendix?
- Meanwhile, from my perspective, the setup in this paper is more close to "online meta-learning" instead of "meta-learning", so it's probably more appropriate to call it "Online Meta-learning in Games".

**Summary Of The Paper:**

This paper proposes and investigates the problem of meta-learning in games. In particular, it builds the framework that uses meta-learning techniques to learn Nash equilibrium sequentially for a set of games by exploiting similarities between these games. Theoretical guarantees under this framework are derived for many classes of games and problems including two-player zero-sum games, potential games, optimal social welfare problems and stackelberg games. It also provides experiments on two Libratus endgames and shows supreme performance of the proposed meta-learning algorithms.

**Summary Of The Review:**

This paper proposes a novel framework to study meta-learning in games and provides extensive results in many classes of problems. However, some assumptions need to be justified and it looks inherently difficult to put everything it covers into the limited number of pages in a clear way.

---

> ### Author Response · Authors · 2022-11-11
> **Response to Reviewer fvHn**
>
> We thank the reviewer for the helpful feedback. Below we address the concerns.
>
> Re. “One major weakness of this paper in my concern is in its organization. Since the coverage is too extensive, the current main content is more like a list of results.”
>
> We understand that the volume of the results can make the paper hard to digest. While the feedback of reviewers xPwr and RKWM in terms of the writing and the organization was positive, please let us know if there is a suggestion on how to improve readability. We have tried to highlight the key algorithmic ideas and the technical difficulties present in each setting, while at the same time introducing enough background without assuming any prior familiarity, so as to make the paper reasonably self-contained.
>
> Re. “most of the theoretical results are derived based on Theorem B.3...”
>
> While Theorem B.3 is the basis for Theorem 3.1 and Theorem 3.6, it is *not* the case that most of our results rely on it. With that in mind, we do not feel that Theorem B.3 deserves more discussion at the expense of discussing specific problems.
>
> Re. “In this paper, the algorithm requires exact Nash equilibrium about previous game.”
>
> We would like to clarify that **only** Theorem 3.2 requires knowledge of the Nash equilibria of previous games; none of our other results makes that assumption. Moreover, as we explain after Theorem 3.2, that guarantee can also be expressed (without any change in the result) in terms of the similarity of the optimal in hindsight as long as the players initialize at the average of the optima-in-hindsight strategies, which does not require any further assumption (as in Theorem 3.1). Also, note that Theorem 3.2 can be directly recast using approximate Nash equilibria for the initialization (which have been learnt by the players without any further assumptions), with an additional term that depends on the approximation error of the Nash equilibria–which vanishes to $0$ as the number of repetitions $m$ increases.
>
> That being said, we argue that assuming access to an exact Nash equilibrium of the previous game (essentially as “side information”) can be justified in settings where players have access to the game after its termination. As we point out after Theorem 3.2, this also relates our paper to the line of work on algorithms with predictions, a connection that we find appealing.
>
> Re. “What is the definition of average potential game?”
>
> For each potential game $t$, let $I(t, \epsilon)$ be the number of iterations required to reach an $\epsilon$-approximate Nash equilibrium. Theorem 3.4 gives a bound on $\frac{1}{T}  \sum_{t=1}^T I(t, \epsilon)$, a quantity which we refer to as the iteration complexity for an average potential game; the same applies to Theorem 3.2.
>
> Re. “It is not very clear what is the "Meta component" in the proposed algorithm.”
>
> The basic idea is that we meta-learn the initialization (and for some of our results other parameters such as the learning rate) of different base learning algorithms commonly used in repeated games. That is, in the most basic version of our algorithm, each player employs an additional regret minimizer that meta-learns the initialization of the optimistic gradient descent algorithm. If the games are similar, according to natural similarity metrics we introduce such as distance between equilibria, initialization using knowledge from previous games can lead to an improvement in performance, and this is exactly what the meta regret minimizer guarantees.
>
> Re. “Is there anything in the algorithm or analysis that captures the similarities among different games?”
>
> Yes, this is one of the central themes of our work. For each setting, we introduce a natural measure that captures the similarity in the sequence of games; for example, in zero-sum games one of the notions of task similarity we employ is the variance of the Nash equilibria across the games (see Subsection 1.1 for a detailed overview). We emphasize, however, that our algorithms are agnostic to the similarity between the games.
>
> Re. “Is it possible to meta-learn these games in a batch fashion like MAML instead of an online fashion?”
>
> We are not sure we have understood what you mean by “batch” here, and would appreciate some clarification. If you mean “batch” in the theory sense, i.e. the question is whether we can PAC-learn initializations when games are drawn from a fixed distribution rather than given by an adversary in an online fashion, then yes: applying online-to-batch conversion (Cesa-Bianchi et al., 2004) to our results would yield such guarantees. If you mean “batch” in the sense of whether there is a minibatch version of our algorithm that can update using multiple games at once, then also yes: similar to the Reptile algorithm (Nichol et al., 2018), at each iteration we can obtain the last-iterates from running the within-task algorithm on each in a batch of B games and then use the average last-iterate to update the meta-initialization.

---

> > ### Comment · Reviewer_fvHn · 2022-11-26
> > **Response**
> >
> > Thank you very much for your detailed response and most of my concerns have been well-addressed! Honestly, I cannot figure out a good way to incorporate such a wide coverage into 9 pages with desired readability, which is the main reason I suggested **a journal submission**.
> >
> > Meanwhile, I think Algorithm 1 and Theorem B.3 deserve more discussion in the main content because from my perspective, it is hard to get the core algorithmic idea without seeing them. And, as a minor comment, it will be better to clarify the "average potential game" since it otherwises sounds like a game obtained by averaging the reward functions of several games.
> >
> > Finally, I would like to keep my original score because I still don't think it is a good fit as a conference paper. However, considering its contribution, I'm also inclined to acceptance.

---

### Official Review · Reviewer_RKWM · 2022-10-24

**Confidence:** 3
**Correctness:** 4
**Technical Novelty And Significance:** 3
**Empirical Novelty And Significance:** 2
**Recommendation:** 8

**Clarity, Quality, Novelty And Reproducibility:**

Overall the paper is clearly written and clear to follow. The theoretical analysis and implementation details are both provided. I was not able to find the implementation code.

**Strength And Weaknesses:**

Strength:
- I found the meta-learning in game perspective interesting, and the analysis and results in the paper directly lead to several problems for future works, such as more general game structures where the single-game results are known but the meta-game analysis remains open.
- The authors were able to consider a set of common game structures and provide a comprehensive set of results. The experiments on the Poker endgames also nicely complement the theoretical analysis.
- Overall the paper is well-written and the structure was very clear to follow.

Weakness:
- in the theoretical guarantees the task similarity notion significantly affects the convergence rate. It would be good to see more interpretation on the "task similarity" notion. What are some examples of sequences of games that exhibit a high/low task similarity?
- In the experiments, how does the proposed algorithm compare with the one in [Zhang et al 2022b]? Given that their solution is for a more general setup that applies in the paper's setting, it would be good to see such comparisons.

**Summary Of The Paper:**

This paper studies a framework on meta-learning in multi-agent games and considered a range of popular game structures, including zero-sum games, potential games, general-sum multi-player games, and Stackelberg security games. The general game is composed by  a sequence of multiple sub-games, where each sub-games have a number of iterations.

The authors analyzed theoretically the rates of convergence to different game equilibria which is dependent on the similarity between the sequence of sub-games, when each of the agents is performing some type of gradient descent updates. Experimentally, the game dynamics were demonstrated with two public Poker endgames.

**Summary Of The Review:**

Overall the paper studies a new framework for meta-learning in several common game structures. The results cover a nice set of game settings and open up a few problems for future works. I suggest the authors to add further interpretations on the theoretical bounds, and experimental comparison with one of the prior works ([Zhang et al 2022b].




-------- post rebuttal --------

I have read the author's response and remain the original score.

---

> ### Author Response · Authors · 2022-11-11
> **Response to Reviewer RKWM**
>
> We thank the reviewer for the helpful feedback.
>
> Re. “It would be good to see more interpretation on the "task similarity" notion. What are some examples of sequences of games that exhibit a high/low task similarity?”
>
> We would like to point out that we do provide examples of sequences of games wherein our notion of task similarity is small: for the similarity of the NE in zero-sum games see below Theorem 3.2; for the similarity of the potential games see below Theorem 3.4; and for the similarity in Stackelberg security games see below Theorem 3.8. Overall, the task similarity is large whenever the sequence of games does not exhibit much structure, e.g., when the sequence of games is random. Let us know if and how we can provide further examples to illustrate our notions of task similarity.
>
> Re. “In the experiments, how does the proposed algorithm compare with the one in [Zhang et al 2022b]? Given that their solution is for a more general setup that applies in the paper's setting, it would be good to see such comparisons.”
>
> We chose not to compare our algorithm experimentally with the algorithm of Zhang et al. because they tackle a more general problem, so we believe that it would not be a fair/meaningful comparison since their algorithm is not tailored to our setting. We refer to Appendix A for a theoretical justification as to why this is the case.

---

### Official Review · Reviewer_K7Pm · 2022-10-26

**Confidence:** 3
**Correctness:** 3
**Technical Novelty And Significance:** 3
**Empirical Novelty And Significance:** 3
**Recommendation:** 8

**Clarity, Quality, Novelty And Reproducibility:**

The paper is novel and of good quality.
The writing in the main part of the paper is confusing in parts.

**Strength And Weaknesses:**



I think this is very good paper. I really have nothing to complain about about the results, the results also are a big enough advance.
The only issue I have is with the writing - I understand that this is a difficult paper to write, in fact I feel this paper is better suited for a journal than a venue which is so much page constrained. I found the informal theorem statements quite very informal, and the formal statement corresponding  to the theorem in the appendix requires tracking a few other theorems in the appendix. I think the appendix is more interesting than the main paper.

Examples of too informal:
In Thm 3.1 it is stated "meta-learning algorithm for the initialization" - which algorithm? - later in the text below it is stated as Alg 1 but it is confusing to state something without introducing it first.
Thm 3.2 does not mention anything about meta-learner (or even the init strategy in the text before the Thm)
In definition of smooth games, OPT is not specified

**Summary Of The Paper:**

This paper introduces meta-learning for equilibrium finding and learning to play games including two-player zero-sum games, general-sum
games, and Stackelberg games. The authors define somewhat natural notions of similarity between the sequence of games encountered. The work evaluates
the meta-learning algorithms on endgames faced by the poker agent Libratus. The experiments show that related games can be solved significantly faster using the meta-learning techniques than by solving them separately.

**Summary Of The Review:**

Solid paper, main paper can be written more clearly.

---

> ### Author Response · Authors · 2022-11-11
> **Response to Reviewer K7Pm**
>
> We thank the reviewer for the helpful suggestions to improve the clarity of the writing. We will make sure to incorporate them in the revised version.

---

### Official Review · Reviewer_xPwr · 2022-11-01

**Confidence:** 4
**Correctness:** 4
**Technical Novelty And Significance:** 3
**Empirical Novelty And Significance:** 2
**Recommendation:** 6

**Clarity, Quality, Novelty And Reproducibility:**

The paper is clearly written and the framework for meta learning in games is clear and easy to understand. Moreover the technical results are all well-argued for and overall the paper has a nice flow to it. I feel the paper is less novel in the sense that it applies a well-known technique in machine learning to learning in games, and the resulting analyses, while interesting, are mostly extensions of known results. Overall, however, the central idea is well motivated and potentially practically useful for future work.


**Strength And Weaknesses:**

Strengths:
- This paper is very thorough in its exposition of related work and in extending the results in each of the settings studied. It is very ambitious to analyze so many different game types and settings, but I feel that the authors managed to do so in a way which still feels digestible and natural.
-  As far as I can tell, despite the breadth of the paper's analyses, the theoretical results are presented clearly and the proofs (while typically derivative of the standard techniques in the field) contain some non-trivial ideas which would certainly be of independent interest in static game settings. A primary example of this is the analysis of the extragradient algorithm using a regret-proxy, allowing for an RVU-type bound to be written even though extragradient is not a no-regret algorithm.

Weaknesses:
- A somewhat minor weakness of the paper is that a lot of the results seem unsurprising, in the sense that by selecting natural similarity metrics and having agents learn initializations based on a 'stacking'-type setup with no-regret algorithms, one would certainly expect the performance to improve over time. It is of course still useful and interesting to formally analyze the performance improvements, but I would have liked to see more focus on the empirical results.
- The experimental results in the paper are quite restricted, in the sense that despite a broad range of theoretical results, the authors only show experimental improvements for zero-sum games. It would have been interesting to see a visual representation of the improvements in potential, general-sum and even stackelberg games using meta-learning.
- Finally, the choice of similarity metric in each setting, while usually appropriately motivated, leaves some room open for debate. In particular it would have been interesting to empirically compare different similarity metrics in settings where multiple metrics can be devised.

**Summary Of The Paper:**

This paper applies the well-known idea of meta learning to learning in games, across various game types and settings including zero-sum games, potential games, general-sum games and Stackelberg games. The authors show several theoretical results for convergence to relevant performance metrics in these games, depending on natural notions of similarity between games encountered by the learner. In general, meta-learning serves to improve the performance of standard algorithms, recovering the known guarantees in static games. The authors also experiment with meta learning poker (zero-sum) endgames, with meta learning used to determine appropriate initializations for the learners. The experiments show that the technique greatly improves performance compared to vanilla methods.


**Summary Of The Review:**

In summary, this paper introduces a framework for meta learning in games which seems to be a useful and interesting idea to me. Many theoretical results in different game settings are described, all of which improve upon vanilla bounds and some new techniques are even shown in the supplementary material. The paper is well written enough to not feel bloated, but there is a lack of empirical evidence for the meta learning techniques in games beyond zero-sum. Overall I would recommend the paper for acceptance.

---

> ### Author Response · Authors · 2022-11-11
> **Response to Reviewer xPwr**
>
> We thank the reviewer for the helpful feedback.
>
> Re. “The experimental results in the paper are quite restricted, in the sense that despite a broad range of theoretical results, the authors only show experimental improvements for zero-sum games...”
>
> While we agree that more experiments across a wider range of game settings could be interesting, we would like to emphasize that we view our theoretical results as the main contribution of our paper. As a result, we decided to focus only on one *real* experimental setting due to space constraints.
>
> Re. “The choice of similarity metric in each setting, while usually appropriately motivated, leaves some room open for debate. In particular it would have been interesting to empirically compare different similarity metrics in settings where multiple metrics can be devised.”
>
> We would like to point out that we do empirically compare the performance of different meta-learning algorithms that are designed to adapt to different notions of task similarity in our experiments: one is based on the distance between optima-in-hindsight (OGD while meta-learning the initialization; Theorem 3.1) and one which depends on the distance between the last iterates (OGD while setting the initialization equal to the last iterate of the previous task; see Remark B.8). Moreover, our illustrative example before Subsection C.1.3 provides an explicit comparison between two different notions of task similarity in zero-sum games, a discussion continued in the paragraph before Remark C.12. Let us know if we can provide additional comparisons.

---

> > ### Comment · Reviewer_xPwr · 2022-11-24
> > **Response**
> >
> > I thank the authors for their response. I do understand that sending this paper to a conference puts some constraints on the writing style, which may not work in the best interest of this particular work since there is quite a lot to cover. Maybe a journal could be a better fit as the last reviewer suggests in terms of optimizing for readability. I still believe that more examples would be valuable here. In any case, as I have said above I am happy to accept the paper as I feel that despite some shortcoming it offers valuable insights.

---

### Decision · Program_Chairs · 2023-01-20

**Decision:**

Accept: poster

**Justification For Why Not Higher Score:**

The meat of the paper is in the supplementary material. Hence, the paper itself could benefit from a major rewriting to improve clarity. At the same time, while the improvement on the numerical example is quite convincing, there are a few overstatements at times, in particular a comparison to [Zhang et al 2022b], which covers the theoretical setting here, is avoided carefully with theoretical justifications. Moreover, the meta-learning aspects are missing (i.e., standard meta-learning aspects are missing).

**Justification For Why Not Lower Score:**

While the results are stated a bit informally, and there is a bit of overreach, the paper still has a solid contribution with a great background review.

**Metareview: Summary, Strengths And Weaknesses:**

The paper works in the meta-learning twist into the online learning in games literature. The main premise is not surprising: If you have a class of similar games, learning from observed games somehow helps with the convergence speed of a new one.

While the writing is a bit informal at times, the overall improvements in practice are significant to warrant publication. The background summary is quite well-done, though there are some additional recent work on weak MVIs that the authors might want to include in camera ready.

While the title is catchy, the paper is making a slight over-reach into the meta-learning field, which covers much more than what the authors use in their paper.

**Note From Pc:**

if the above contains the word "oral" or "spotlight" please see: "oral" presentation means -> notable-top-5% and "spotlight" means -> notable-top-25%. As stated in our emails, we are disassociating presentation type from AC recommendations